# Incidence, Risk Factors, and Consequences of Post-Traumatic Stress Disorder Symptoms in Survivors of COVID-19-Related ARDS

**DOI:** 10.3390/ijerph20085504

**Published:** 2023-04-13

**Authors:** Sara Miori, Andrea Sanna, Sergio Lassola, Erica Cicolini, Roberto Zanella, Sandra Magnoni, Silvia De Rosa, Giacomo Bellani, Michele Umbrello

**Affiliations:** 1Department of Anesthesia and Intensive Care, Santa Chiara Hospital, 38122 Trento, Italy; 2Centre for Medical Sciences—CISMed, University of Trento, 38122 Trento, Italy; 3Department of Intensive Care Unit, San Carlo Borromeo University Hospital, 20142 Milan, Italy

**Keywords:** COVID-19, acute respiratory distress syndrome, post-traumatic stress disorder, critical care, follow up, quality of life

## Abstract

**Purpose:** To assess the prevalence of symptoms of Post-Traumatic Stress Disorder (PTSD) in survivors of COVID-19 Acute Respiratory Distress Syndrome that needed ICU care; to investigate risk factors and their impact on the Health-Related Quality of life (HR-QoL). **Materials and Methods:** This multicenter, prospective, observational study included all patients who were discharged from the ICU. Patients were administered the European Quality of Life 5 Dimensions 5 Level Version (EQ-5D-5L) questionnaire, the Short-Form Health Survey 36Version 2 (SF-36v2), a socioeconomic question set and the Impact of Event Scale—Revised (IES-R) to assess PTSD. **Results:** The multivariate logistic regression model found that an International Standard Classification of Education Score (ISCED) higher than 2 (OR 3.42 (95% CI 1.28–9.85)), monthly income less than EUR 1500 (OR 0.36 (95% CI 0.13–0.97)), and more than two comorbidities (OR 4.62 (95% CI 1.33–16.88)) are risk factors for developing PTSD symptoms. Patients with PTSD symptoms are more likely to present a worsening in their quality of life as assessed by EQ-5D-5L and SF-36 scales. **Conclusion:** The main factors associated with the development of PTSD-related symptoms were a higher education level, a lower monthly income, and more than two comorbidities. Patients who developed symptoms of PTSD reported a significantly lower Health-Related Quality of life as compared to patients without PTSD. Future research areas should be oriented toward recognizing potential psychosocial and psychopathological variables capable of influencing the quality of life of patients discharged from the intensive care unit to better recognize the prognosis and longtime effects of diseases.

## 1. Introduction

Among the long-term effects observed in survivors after COVID-19, with or without hospitalization, there are consequences on physical and mental health that often compromise the daily functioning and quality of life of patients and their families [1,2,3]. As the concept of health, according to the World Health Organization (WHO), should not be considered merely the absence of disease or infirmity, but a state of complete physical, mental, and social well-being [4], it is of utmost importance to study the prevalence and determinants of health problems during the COVID-19 pandemic.

Several studies have reported a substantial increase in mental health challenges among survivors [5,6,7], but also in caregivers [3] and frontline healthcare workers [8,9], exposing public health to great challenges [10]. Mental health problems after intensive care unit (ICU) discharge during the COVID-19 pandemic are particularly numerous and disabling [11,12], possibly as a consequence of exposure to extremely stressful acute events and particularly severe forms of disease, besides the often very long hospital stay and the incidence of post-intensive-care syndrome [6,7,13,14]. 

Psychiatric symptoms can include anxiety, depression, and post-traumatic stress disorder (PTSD). The latter is a highly disabling psychopathology frequently experienced by people who are exposed to natural and human-made disasters, mass and war violence, terrorism, or child abuse, but high rates are also registered after ICU stays [14,15,16], as well as in COVID-19 survivors and their families [17,18,19]. The last edition of the Diagnostic and Statistical Manual of mental disorders (DSM-V) defines PTSD as a mental health condition characterized by intrusive memories; symptoms of avoidance of places, activities, or people that remind individuals of the traumatic event; negative changes in thinking and moods [20], and many instruments have been studied to investigate its presence. Despite this, it is questionable if PTSD is the proper diagnosis after the COVID-19 pandemic according to DSM-V’s inclusion criteria, especially regarding the direct experience (Criterion A1) or witness (Criterion A2) of a traumatic event, including actual or threatened death, and serious injury [21]; indeed, if we cannot consider every disease (e.g., the flu) as a life-threatening situation, some authors have stated that an adjustment disorder (AD) might be a more suitable diagnosis [22], given that the ICD-11 definition of adjustment disorder includes serious diseases [23].

Recent studies identified PTSD as a major health issue caused by COVID-19 infection, and survivors after an ICU stay are at higher risk of developing this condition, as compared to the population not exposed to the most severe forms of the disease or managed at home without any ventilatory support [17,24,25]. Risk factors related to the development of PTSD after ICU discharge include the length of hospital and ICU stays, mechanical ventilation, disease severity, higher levels of PCR, or lower PaO_2_/FiO_2_ ratio [18]. However, the role of social determinants such as social support, family composition, monthly income, and level of education has never been formally investigated, though the literature has emphasized their role in mental health and resilience after disasters [26]. 

The aim of the current study was to examine the prevalence of symptoms of PTSD in survivors of COVID-19 acute respiratory distress syndrome (C-ARDS) that needed ICU care, to investigate clinical risk factors associated with the development of such symptoms and to address whether social determinants could have a significant role in its incidence. The secondary aim was to assess whether the development of symptoms of PTSD has an impact on the Health-Related Quality of Life (HR-QoL) of C-ARDS survivors.

## 2. Materials and Methods

### 2.1. Study Design and Participants

This multicenter, prospective, observational study included all patients who were discharged from the ICU after admission for C-ARDS, defined according to the Berlin criteria [27], in two distinct referral hospitals in the North-East Italy region Trentino-Alto Adige (S. Chiara Hospital -Trento and S. Maria del Carmine Hospital—Rovereto) between March 2020 and September 2021. All adult (>18 years), Italian-speaking patients who required ICU assistance for more than 48 h were eligible for inclusion. Non-Italian-speaking patients or those who denied informed consent were excluded. 

Six months after hospital discharge, COVID-19 ICU survivors were first contacted by an experienced ICU physician previously involved in patient clinical management (AS, SM, EC) with an informative phone call, explaining the purpose of the study and methods of data processing; then, the patients were contacted again with a second phone call, only if they had sent by email a signed informed consent form. Those who did not respond at follow-up or denied participating were excluded. A flow chart of the study is shown in Figure 1. 

During the structured telephone interview, patients were administered a set of questionnaires including: the five-level, five-dimensional EuroQoL questionnaire (EQ-5D-5L) [28], the EuroQol Visual Analog Scale (EQ VAS) Short Form, and the Short-Form Health Survey 36Version 2 (SF-36v2) [29] to assess the HR-QoL; a question set designed to investigate the socioeconomic status and changes in family and work circumstances and care needs [12]; the Italian version of the Impact of Event Scale—Revised (IES-R) to assess PTSD [30]. Telephone interviews lasted an average of 45 min.

The study was approved by the Ethics Committee of Azienda Provinciale per i Servizi Sanitari di Trento (APSS; Protocol number 1/2021) as part of a large follow-up study on COVID-19 survivors. 

All participants provided a written informed consent before answering the questionnaires. 

### 2.2. Instruments and Data Collected

Clinical measures: Clinical information about ICU stay, laboratory data, and severity scores was digitally recorded and extracted using the Institutional electronic medical record. We considered “highest value” as the worst data during the ICU stay. All data were anonymized and saved in an electronic worksheet.IES-R: The presence of PTSD symptoms was assessed with the Italian version of IES-R, which is a 22-item-self-report and easily administered questionnaire that assesses subjective distress caused by traumatic events. Items correspond directly to 14 of the 17 criteria of the Diagnostic and Statistical Manual of Mental Disorders, Fourth Edition (DSM-IV). Patients are asked to identify a specific stressful life event and then rate how much they were distressed or bothered by the difficulty during the previous 7 days, e.g., “Any reminders brought back feelings about it”. Items are rated on a 5-point scale ranging from 0 (“not at all”) to 4 (“extremely”). Item scores are summed to give the total score (ranging from 0 to 88). Subscales can also be calculated for Intrusion, Avoidance, and Hyperarousal. A total IES-R score of >33 out of a theoretical maximum of 88 implies the likely presence of PTSD, although the IES-R scale is not used to give a formal diagnosis of PTSD.EQ-5D-5L and EQ VAS scale: The former describes 5 dimensions of health, i.e., mobility, self-care, usual activities, pain/discomfort, and anxiety/depression, with five levels each: 1. no problems, 2. slight problems, 3. moderate problems, 4. severe problems, and 5. extreme problems. The patient is asked to indicate his/her health state in each of the 5 dimensions. The latter records the self-rated health condition on a vertical, visual analog scale where the two extremes are ‘Best imaginable health state’ and ‘Worst imaginable health state’.SF-36 Version 2: This is an 11-question, 36-item questionnaire frequently used to describe the HR-QoL after ICU stays, which investigates 8 health domains, i.e., physical functioning, physical role, pain, general health, vitality, social functioning, emotional role, and mental health. Domain scores and Physical (PCS) and Mental (MCS) Component Summary scores were calculated as recommended in the specific manual and interpretation guide [31]. The overall score on each SF-36 subscale ranges from 0 to 100 with higher scores indicating a better HR-QoL.Sociodemographic variables: All participants were administered a multiple-choice questionnaire to evaluate the socioeconomic impact of COVID-19; age, gender, marital status, family composition, municipality of residence, education level (using the International Standard Classification of Education—ISCED—ISCED 0: Early childhood education (‘less than primary’ for educational attainment), ISCED 1: Primary education, ISCED 2: Lower secondary education, ISCED 3: Upper secondary education, ISCED 4: Post-secondary non-tertiary education, ISCED 5: Short-cycle tertiary education, ISCED 6: Bachelor’s or equivalent level, ISCED 7: Master’s or equivalent level, ISCED 8: Doctoral or equivalent level), occupation (full time, part time, unemployed, retired), salary level, and the eventual presence of a possible stigmatization process of the survivor were included.

### 2.3. Statistical Analysis

Continuous variables are reported as means (standard deviation) if normally distributed or medians (25th; 75th centile) if not; categorical variables are shown as numbers and percentages. The variable distribution was tested by the Kolmogorov–Smirnov and Shapiro–Wilk tests. Subjects were divided in groups according to the development of symptoms of PTSD, defined as an IES-R score of >33. Continuous variables were compared with appropriate parametric or non-parametric tests according to their distribution, and categorical variables were compared with chi-square tests.

Social and disease-related risk factors and clinically and epidemiologically relevant variables were selected a priori and dichotomized using meaningful reference values, when applicable. Univariate binomial logistic regression analysis was used to explore risk factors (covariates) associated with the development of symptoms of PTSD (dependent variable). A multivariable binomial logistic regression analysis was used to explore associated factors with the development of symptoms of PTSD. For model selection, a forward stepwise procedure based on the Akaike information criterion (AIC) was used; adjusted odds ratios with 95% confidence intervals for the variables included in the model were calculated. The discriminative ability of the final model was assessed with the area under the receiver operating characteristic (ROC) curve, and the Hosmer–Lemeshow goodness-of-fit statistics was used to assess the calibration.

All data were analyzed using Statistical Package for Social Science version 20 (IBM Corp, Armonk, NY, USA) or Microsoft Excel 2020 (Microsoft Corporation, Redmond, WA, USA). For all the analyses, *p*-values < 0.05 were considered statistically significant. 

## 3. Results

During the study period, 142 patients were recruited at the 6-month follow-up; 34 patients (23.9%) developed Post-Traumatic Stress Disorder symptoms. Table 1 shows demographic, clinical, and socioeconomic characteristics of the whole case-mix and of two subgroups with or without PTSD symptoms. 

The multivariate logistic regression model found that an ISCED score higher than 2 (OR 3.42 (95% CI 1.28–9.85)), monthly income less than EUR 1500 (OR 0.36 (95% CI 0.13–0.97)), and the presence of more than two comorbidities (OR 4.62 (95% CI 1.33–16.88)) are significant risk factors for developing PTSD symptoms. The area under the ROC curve of the model was 0.74 and the Hosmer–Lemeshow goodness of fit for the logistic regression model showed *p* = 0.78. Figure 2 shows the variables included in the multivariate logistic regression model.

Figure 3 and Appendix A report the comparison of the distribution of the five EQ5D5L dimensions in patients who did vs. those who did not develop symptoms of PTSD. Significant differences were found between groups in 3 out of 5 dimensions (anxiety and depression, mobility, and usual activities, with patients who developed PTSD who reported higher levels of impairment in their quality of life. Figure 4 and Appendix A show the results of the SF-36 questionnaire, with a significant reduction in each area, except role limitations due to physical health. Similarly, EQ-VAS (74.8 ± 16.0 vs. 66.0 ± 17.5, *p* = 0.008), as well as both the Physical (44.8 ± 10.3 vs. 40.3 ± 10.5, *p* = 0.028) and the Mental Component Score (48.4 ± 8.3 vs. 40.3 ± 12.3, *p* < 0.001), are significantly higher in patients who did not develop symptoms of PTSD.

## 4. Discussion

To the best of our knowledge, this is the first observational, follow-up study on COVID-19 ICU survivors that investigated the association of both clinical and socioeconomic risk factors with the onset of symptoms related to Post-Traumatic Stress Disorder. The psychological consequences of collective traumatic events such as hurricanes [32], earthquakes [33], or terrorist attacks [34] have been the subject of study for many years, but natural disaster or terrorist attack characteristics are extremely different from a pandemic outbreak, albeit unexpected and serious, and for this reason, COVID-19 infection mental health effects [7,14,35,36] must be considered a separate case. Indeed, medical conditions from natural causes such as life-threatening viral infection do not meet the current criteria for trauma required for a diagnosis of PTSD, but other psychopathologies, such as depressive, anxiety, or adjustment disorders, may ensure [21,37].

A stay in the ICU is itself a traumatic experience, as patients are confronted with their own death, are dependent on machines, and might be altered in consciousness, with severely impaired or impossible communication skills [38]. In particular, in our study, all patients presenting with PTSD symptoms recall hospitalization as a traumatic experience, but only a fraction of them were able to recall a specific traumatic event related to their ICU stay, probably due to the use of sedatives and early discharge to intermediate-care wards as soon as possible because of ICU bed shortages.

The incidence of PTSD in ICU survivors is of particular concern due to the frequently overlapped post-intensive-care syndrome [6,24,39], which severely complicates recovery after severe illness. This is of further importance, as we found an incidence more than twice that reported in the general literature, as the incidence of PTSD in the first year after hospital discharge, as the sole consequence of the recent ICU discharge and not related to past stressful experiences, has been consistently reported around 10% in the most rigorous studies [40,41,42]. Our results are comparable with other follow-up studies in patients hospitalized for COVID-19 [7] and this confirms the IES-R scale as a useful tool to identify the symptoms of a potential PTSD.

Predictors of the onset of PTSD symptoms after hospital or ICU discharge are not clearly defined and vary in different studies; some authors could not find any clear correlation between disease severity and PTSD [43], while others identified as risk factors traumatic memories during ICU stays, the duration of sedation, opioid dosage, nightmares, and feeling breathless [44,45]; or lower education level, alcohol abuse, and female sex [43]. The COVID-19 pandemic exposed patients and healthcare workers to emotional distress and increased the risk for psychiatric illness due to some specifics such as uncertain prognoses, severe shortages of resources, the imposition of unfamiliar public health measures, large financial losses, and conflicting messages from authorities [21]. Janiri et al. [7] found a positive correlation between persistent mental health symptoms and female sex, a history of psychiatric disorders, and delirium or agitation during acute illness. Ju et al. [36] stated that patients with lower educational levels, higher anxiety levels, and lower perceptions of emotional support during hospitalization may be more likely to develop PTSD. Tarsitani et al. [45] identified previous psychiatric history and obesity as risk factors. However, other studies observed contrasting results [46,47,48], and this confirms that the psychological susceptibility to COVID-19 consequences is still not well known.

Socioeconomic status is a well-known essential factor associated with a greater risk of psychopathological issues after disasters [49,50,51]. In the multivariate analysis, we found that a monthly income less than 1500 euros is a predisposing factor to PTSD symptoms. At the moment, no study has ever investigated the income level as a predisposing factor of PTSD in the COVID-19 era and, for this reason, we are not able to provide a comparison with other studies, but PTSD probability tends to be greater among more economically vulnerable people [52,53] and COVID-19 might not be an exception.

Furthermore, female gender was shown to be a risk factor for the onset of symptoms related to PTSD. This finding is in line with numerous post-COVID-19 studies [54,55,56] and confirms the reduced ability of the female gender to deal with traumatic events, due to psychosocial and biological explanations (e.g., oxytocin related) [57]. In the multivariate analysis, the female gender-role was not confirmed as a risk factor for the development of symptoms related to PTSD; this is in contrast with previous literature concerning stressor events, but it could be the sign of a different trajectory in the development of PTSD symptoms, not related to gender, but to a general greater sensitivity to mental health problems even in traditionally excluded parts of the population (i.e., the older men) due to cultural taboos that do not consider it acceptable to talk about problems in managing traumatic events. Large and stressful media coverage on COVID-19 may have played a significant role [47,58,59,60], but any definitive conclusions will require gender- and sex-sensitive research and reporting.

Quite surprisingly, our study showed that high educational level (ISCED > 2) and low monthly income were significantly associated with an increased risk of PTSD. This is in contrast with other studies on trauma-exposed adults [61] or after natural disasters [62] and contradicts the theory that education allows the acquisition of effective tools to face threats and traumatic events [63,64]. This result could have two plausible explanations. First, it is a probable effect of the social composition of the study area, mainly composed of high-income subjects with low educational level, employed in rural and mountain tourism activities, and people with high educational level but low monthly income. This particular social situation could have amplified the susceptibility of the more educated subjects to have mental health problems toward a stressful event, maybe due to a better perception of health-related quality of life changes during the COVID-19 pandemic, without the protective effect of a high salary. Our result is consistent with the finding of higher levels of distress in retired people with higher levels of education [65] and many scientific reports of an alarming percentage of higher-education students with mental health problems after the COVID-19 pandemic outbreak [66,67,68,69], despite a theoretical greater resilience capacity in the more educated people. Factors related to the ability to manage traumatic events are indeed linked not only to educational characteristics, but also to material living conditions related to income, housing conditions, and general wealth or socio-relational resources. Secondly, our study could be a further example of the serious methodological issues related to research on PTSD in the context of the COVID-19 pandemic [70]. It is uncertain if DSM-V’s inclusion criteria for PTSD are compatible with the situations that often occur during the COVID-19 pandemic, as not every disease is a life-threatening event, at least for every patient affected [21,37]. If, as some authors stated [22], adjustment disorder is a more suitable diagnosis because it does not require a stressor that is life-threatening or traumatic [71], it is not surprising that COVID-19 survivors show different risk or protective factors compared to natural-disaster- or trauma-exposed adults.

On the other hand, we did not find correlations with the municipality of residence, likely because the public health system of the territory in which our population lives is universal, homogeneously distributed, and easy to access for anyone, thus reducing the difficulties of access to health services to those who do not live in the main inhabited centers.

The quality of life, as measured by EQ5D5L and SF36 scales, is significantly lower in patients with PTSD-related symptoms, and this is a further confirmation that psychological distress after COVID-19 has a severe impact on the daily life of patients after ICU and hospital discharge [72].

Eventually, we found a high percentage of survivors who complain of a worsening of their social relationships, mainly from acquaintances and friends and much less from close family members. This could reflect a process of stigmatization of the patient, which is known to have been observed under COVID-19 and could be a source of suffering and inequality.

## 5. Limitations

The small sample of our study could limit the possibility to generalize the results obtained. In addition, the IES-R scale is not a suitable tool to provide a formal diagnosis of PTSD, which would require a specialist clinical evaluation, and the DSM-V definition of PTSD is not completely appropriate after a pandemic; PTSD might not be the best clinical diagnosis, when compared, for example, with adjustment disorder. It would have been interesting to use a neurocognitive tool for screening cognitive deficits or a previous psychiatric history for elderly (>60 years) patients in a presential additional session, not possible at that time due to the social restriction imposed. Eventually, the socioeconomic questionnaire evaluated only some macro areas of interest, possibly leaving out other socio-relational and economic areas of patients’ lives that could be significant in the onset of post-traumatic stress disorder symptoms. Future studies could investigate multiple socioeconomic factors to predict PTSD.

## 6. Conclusions

In conclusion, we found how the incidence of PTSD is high at 6 months after hospital discharge in survivors of COVID-19 ARDS; the main factors associated with the development of symptoms of PTSD were socioeconomic, such as a higher education level, a lower monthly income, and the presence of more than two comorbidities. Patients who developed symptoms of PTSD reported a significantly lower HR-QoL as compared to patients without PTSD. Future research areas should be oriented toward recognizing potential psychosocial and psychopathological variables capable of influencing the quality of life of patients discharged from intensive care unit to better recognize prognoses and longtime effects of diseases.

## Figures and Tables

**Figure 1 ijerph-20-05504-f001:**
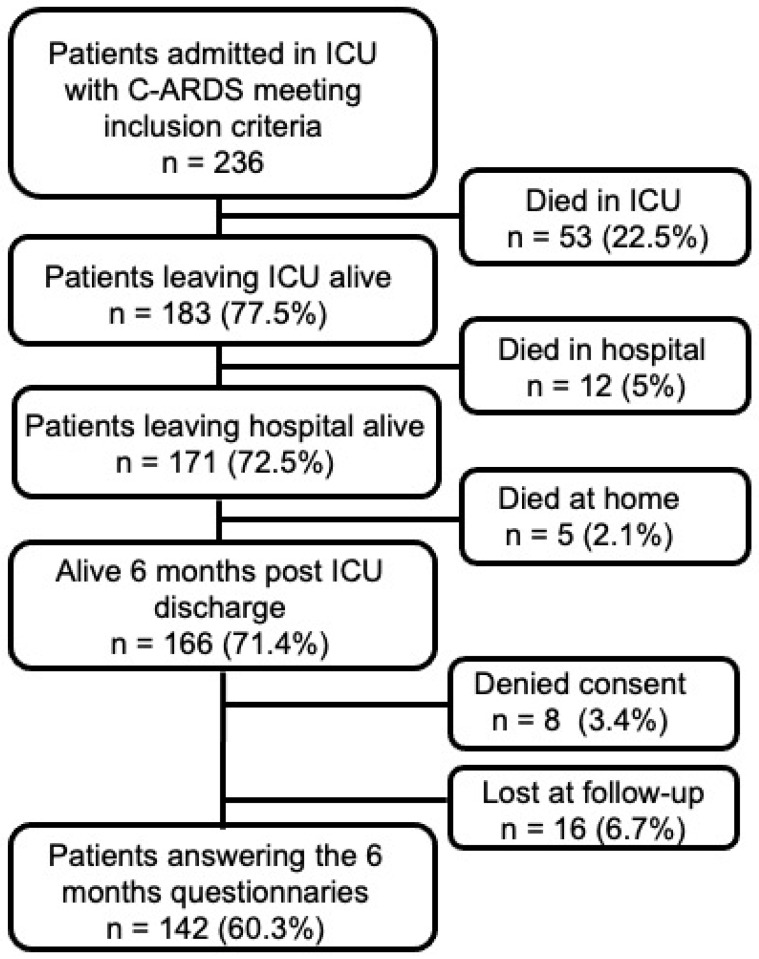
Study flow chart.

**Figure 2 ijerph-20-05504-f002:**
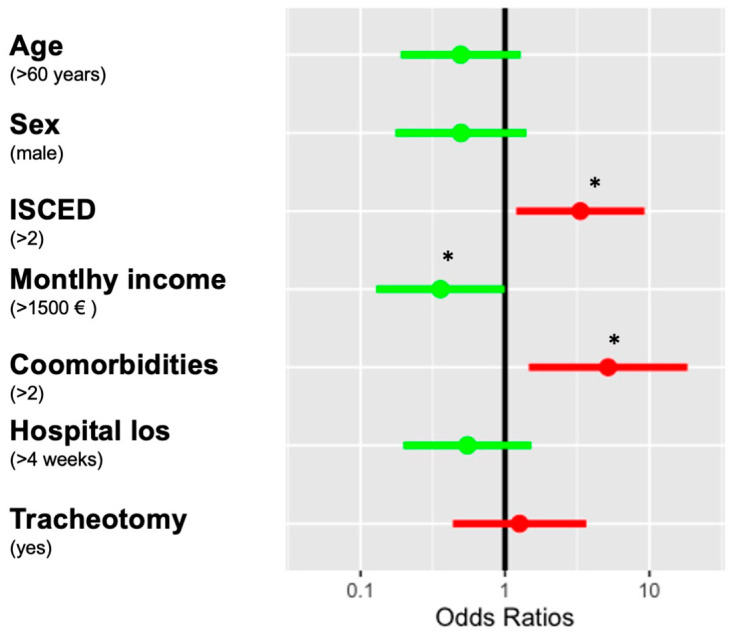
Multivariate logistic regression model predicting the development PTSD-related symptoms. Note: Multivariable analysis of factors associated with the development of PTSD-related symptoms in C-ARDS survivors. The figure shows factors associated with IES-R >33 (dependent variable), the calculated adjusted odds ratios, and the 95% confidence intervals, based on the logistic regression, on the *x*-axis, and on the correspondent forest plot. Red lines show the variables favoring the development of PTSD symptoms and green lines show the protective ones. * *p* < 0.05.

**Figure 3 ijerph-20-05504-f003:**
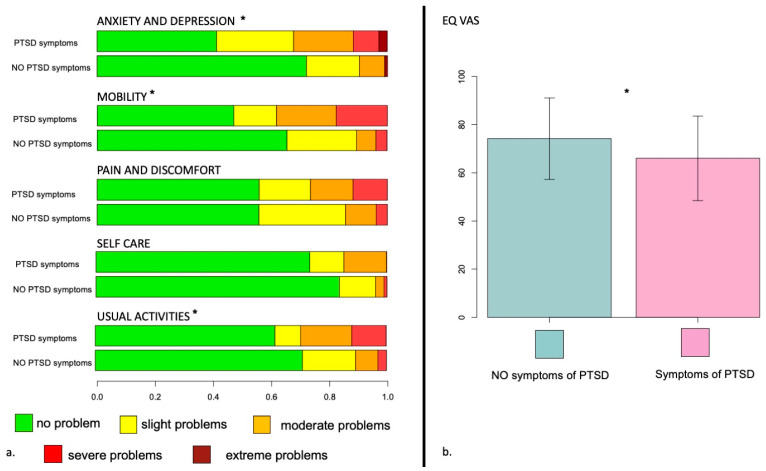
Impact of PTSD-related symptoms on health-related quality of life assessed by EQ-5D-5L questionnaire. Note: (**a**) Frequency distribution of the EQ-5D-5L scores in each of the five domains (pain or discomfort, mobility, usual activities, anxiety and depression, self-care) in C-ARDS survivors’ comparison between patients with and without PTSD-related symptoms. Each domain is scored on a 5-point scale: 1, no problems; 2, slight problems; 3, moderate problems; 4, severe problems; 5, unable to do. (**b**) EQ VAS comparison in patients with and without PTSD-related symptoms. * *p* < 0.05.

**Figure 4 ijerph-20-05504-f004:**
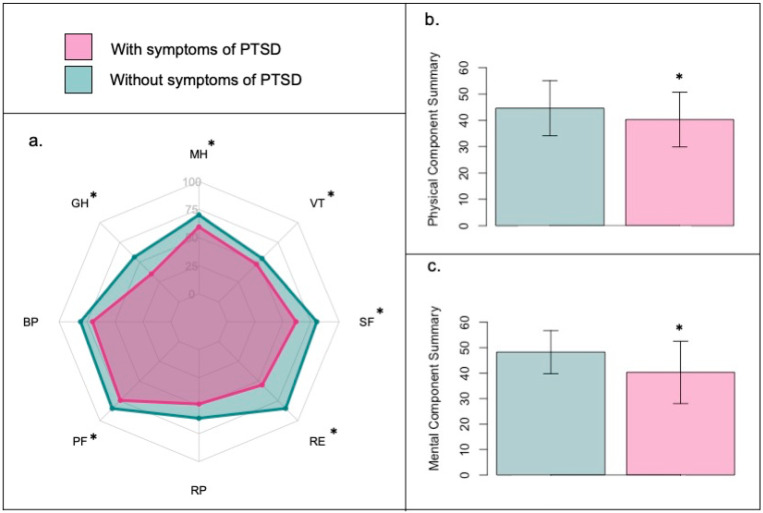
Impact of PTSD-related symptoms on health-related quality of life assessed by SF36 questionnaire. Note: (**a**). Mean scores in SF-36 questionnaire for patients with and without PTSD-related symptoms. PF, physical functioning; SF, social functioning; RP, role limitation due to physical problems; RE, role limitation due to emotional problems; MH, mental health; BP, bodily pain; VT, vitality; GH, general health. (**b**). Physical Component Summary score comparison in patients with and without PTSD-related symptoms. (**c**). Mental Component Summary score comparison in patients with and without PTSD-related symptoms * *p* < 0.05.

**Table 1 ijerph-20-05504-t001:** Demographic, Clinical, and Socioeconomic Characteristics of patients with or without Post-Traumatic Stress Disorder (PTSD) symptoms.

Demographic Characteristics		Whole Sample (*n* = 142)	No PTSD (*n* = 108)	PTSD (*n* = 34)	*p*-Value ^a^
**Age (years)**		63 ± 9	64 ± 9	62 ± 10	0.242
Age, *n* (%)	<50 years	10 (7%)	7 (6.5%)	3 (8.8%)	0.871
	50–60 years	37 (26.1%)	29 (26.9%)	8 (23.5%)	
	60–70 years	61 (43%)	45 (41.7%)	16 (47.1%)	
	>70 years	34 (23.9)	27 (25%)	7 (20.6%)	
Sex. Male, *n* (%)		117 (82.4)	94 (87)	23 (67.6)	0.009
Female, *n* (%)		25 (17.6)	14 (13%)	11 (32.4)	
**Clinical characteristics**					
No preexisting comorbidities, *n* (%)		64 (45.1)	53 (49.1)	11 (32.4)	0.011
One or two comorbidities, *n* (%)		64 (45.1)	49 (45.4)	15 (44.1)	
More than two comorbidities, *n* (%)		14 (9.8)	6 (5.6)	8 (23.5)	
BMI (kg/m^2^)		28 [26–31]	28 [25–31]	28 [26–31]	0.624
Tracheal intubation, days (%)		103 (72.5)	82 (75.9)	21 (61.8)	0.125
Tracheostomy, *n* (%)		44 (30.8)	34 (31.5)	10 (29.4)	>0.999
P/F ratio on ICU admission		158 [108–207]	163 [112–212]	138 [106–173]	0.163
Pronation, *n* (%)		89 (63.6)	67 (62.6)	22 (66.7)	0.836
Steroids, *n* (%)		98 (69.5)	72 (66.7)	26 (78.8)	0.204
ICU LOS, days		18 [10–29]	19 [11–31]	17 [8–23]	0.285
Hospital LOS, days		34 [24–47]	36 [25–47]	33 [22–46]	0.378
**Socioeconomic characteristics**					
Marital status					
Celibate/maiden, n (%)		17 (12)	13 (12.5)	4 (11.8)	0.856
Married, n (%)		106 (75)	77 (74)	25 (73.5)	
Cohabitant, n (%)		7 (5)	5 (4.8)	2 (5.9)	
Divorced, n (%)		2 (1)	2 (1.9)	0	
Separated, n (%)		4 (3)	2 (1.9)	2 (5.9)	
Widow, n (%)		6(4)	5 (4.8)	1 (2.9)	
Family unit composition					
Single, n (%)		18 (12.3)	13 (12)	5 (14.7)	0.768
More than 2 people, n (%)		124 (87.7)	95 (88)	29 (85.3)	
**Population of the municipality of residence**					
<15,000, *n* (%)		97 (68.3)	71 (65.7)	26 (76.5)	0.434
15–50,000, *n* (%)		12 (8.5)	9 (8.3)	3 (8.8)	
>100,000, *n* (%)		33 (23.21)	28 (25.9)	5 (14.7)	
Employment status					
Active worker		51 (35.9%)	39 (36.1%)	12 (35.3%)	>0.999
Retired/unemployed/not working		91 (64.1%)	69 (63.9%)	22 (64.7%)	
**Monthly Income (EUR)**					
<1500		68 (48.6%)	47 (43.9%)	21 (63.6%)	0.072
>1500		72 (51.4%)	60 (56.1%)	12 (36.4%)	
**Education**					
ISCED 0–2		61 (43%)	51 (47.2%)	10 (29.4%)	0.076
ISCED > 2		51 (57%)	57 (52.8%)	24 (70.6)	
**Socio-relational impact**					
Need for assistance in daily life		28 (20%)	23 (21.7%)	5 (14.7%)	0.465
Worsening of social life after hospital discharge		37 (26.4%)	24 (22.6%)	13 (38.2%)	0.079

Note: Data are shown as median (interquartile range), *n* (%), or mean ± SD. Abbreviations: PTSD, post-traumatic stress disorder; BMI, body mass index; P/F ratio, ratio of arterial oxygen partial pressure (PaO2 in mmHg) to fractional inspired oxygen; ICU, intensive care unit; LOS, length of stay; *ISCED*, International Standard Classification of Education; *ISCED 0*, early childhood education (“less than primary” for educational attainment); *ISCED 1*, primary education; *ISCED 2*, lower secondary education; *ISCED 3*, upper secondary education; *ISCED 4*, post-secondary non-tertiary education; *ISCED 5*, short-cycle tertiary education; *ISCED 6*, bachelor’s or equivalent level; *ISCED 7*, master’s or equivalent level; *ISCED 8*, doctoral or equivalent level. ^a^ Comparison between patients with PTSD and without PTSD.

## Data Availability

The data presented in this study are available on request from the corresponding author. The data are not publicly available due to privacy and ethical restrictions.

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
