# Peer review of "Incidence, Risk Factors, and Consequences of Post-Traumatic Stress Disorder Symptoms in Survivors of COVID-19-Related ARDS"

_ijerph, 2023, doi:10.3390/ijerph20085504_

Round 1
Reviewer 1 Report
I liked this study and enjoyed reading it. I have a major problem with it, but I think that just discussing it in the discussion will be enough.
“The psychological consequences of COVID-19 infection [29][14][7][30] and other collective 215 traumatic events such as hurricanes [31], earthquakes [32] or terrorist attacks [33] have been the subject of study for many years.”
This is simply not a comparison that you can make. I know other authors have made the comparison between COVID and terrorism, but this is simply conceptually inane and redundant. One cannot at the one hand claim “COVID is unprecedented” and then compare it to disasters. Secondly, the impact of natural and man-made disasters is clearly different: COVID itself is a disease. If you compare a COVID infection with a terrorist attack, then why not also the flu? In my country there is now a sharp increase of flu, leading to an overburdening of the hospitals. Is the flu suddenly like terrorism?
Secondly, how is a COVID infection a traumatic event? Unless one is threatened with death… DSM-IV had serious diseases as possible trauma, the DSM-V clearly does not. So, while the authors do use the DSM-IV version, they should wonder: why has the DSM-V removed disease as a possible trauma? Because it did not make sense. The ICD-11 definition of adjustment disorder now even includes serious disease.
I would suggest that the authors discuss the difficulty of such methodological issues. The paper by Pfefferbaum and North, who themselves have done a lot of PTSD studies during COVID, also acknowledged that COVID actually does not fit the criteria.
Pfefferbaum B, North CS. Mental Health and the Covid-19 Pandemic. N Engl J Med. 2020 Aug 6;383(6):510-512. doi: 10.1056/NEJMp2008017. Epub 2020 Apr 13. PMID: 32283003.
Asmundson GJG, Taylor S. Garbage in, garbage out: The tenuous state of research on PTSD in the context of the COVID-19 pandemic and infodemic. J Anxiety Disord. 2021 Mar;78:102368. doi: 10.1016/j.janxdis.2021.102368. Epub 2021 Feb 8. PMID: 33582405; PMCID: PMC9759101.
Van Overmeire R. Comment on Dutheil, Mondillon, and Navel (2020): the importance of adjustment disorders and resilience. Psychol Med. 2020 Sep 8:1-2. doi: 10.1017/S003329172000344X. Epub ahead of print. PMID: 32895089; PMCID: PMC7487745.
“Quite surprisingly our study showed that high educational level (ISCED > 2) and low 262 monthly income were significantly associated with increased risk of PTSD. This is in con- 263 trast with other studies on trauma exposed adults [58] or after natural disasters [59] and 264 contradicts the theory that education allows to acquire effective tools to face threats and 265 traumatic events [60][61]”
Again, if you compare it to “other disasters”, then you indeed get strange results.
“A stay in ICU is itself a traumatic experience, since patients are confronted with their own death, are dependent on machines and might be altered in consciousness, with severely impaired or impossible communication skills [34].” This reference here is not very convincing. It is not a study, but a spotlight, that does not even use references. The author of the reference does not really give a reason for why it might be traumatic – I agree that ICU might be traumatic, but it is not the ICU itself, it is what happens to the person in the ICU
Related to that: how many people mentioned that the ICU was traumatic for them? Because in answering the PTSD scale they needed to mention a specific trauma to which it is related, to fulfill criterion A.
I would really suggest the authors simply compare it to other COVID-19 studies – there are so many.
In limitations I would add: 1, use of DSM-IV, not the most current use of PTSD; 2, PTSD might not be the best choice of disorder, when compared to adjustment disorders for example.
Author Response
- I liked this study and enjoyed reading it. I have a major problem with it, but I think that just discussing it in the discussion will be enough.
“The psychological consequences of COVID-19 infection [29][14][7][30] and other collective 215 traumatic events such as hurricanes [31], earthquakes [32] or terrorist attacks [33] have been the subject of study for many years.”
This is simply not a comparison that you can make. I know other authors have made the comparison between COVID and terrorism, but this is simply conceptually inane and redundant. One cannot at the one hand claim “COVID is unprecedented” and then compare it to disasters. Secondly, the impact of natural and man-made disasters is clearly different: COVID itself is a disease. If you compare a COVID infection with a terrorist attack, then why not also the flu? In my country there is now a sharp increase of flu, leading to an overburdening of the hospitals. Is the flu suddenly like terrorism? Secondly, how is a COVID infection a traumatic event? Unless one is threatened with death… DSM-IV had serious diseases as possible trauma, the DSM-V clearly does not. So, while the authors do use the DSM-IV version, they should wonder: why has the DSM-V removed disease as a possible trauma? Because it did not make sense. The ICD-11 definition of adjustment disorder now even includes serious disease.
I would suggest that the authors discuss the difficulty of such methodological issues. The paper by Pfefferbaum and North, who themselves have done a lot of PTSD studies during COVID, also acknowledged that COVID actually does not fit the criteria.
Pfefferbaum B, North CS. Mental Health and the Covid-19 Pandemic. N Engl J Med. 2020 Aug 6;383(6):510-512. doi: 10.1056/NEJMp2008017. Epub 2020 Apr 13. PMID: 32283003.
Asmundson GJG, Taylor S. Garbage in, garbage out: The tenuous state of research on PTSD in the context of the COVID-19 pandemic and infodemic. J Anxiety Disord. 2021 Mar;78:102368. doi: 10.1016/j.janxdis.2021.102368. Epub 2021 Feb 8. PMID: 33582405; PMCID: PMC9759101.
Van Overmeire R. Comment on Dutheil, Mondillon, and Navel (2020): the importance of adjustment disorders and resilience. Psychol Med. 2020 Sep 8:1-2. doi: 10.1017/S003329172000344X. Epub ahead of print. PMID: 32895089; PMCID: PMC7487745.
Response: We thank the reviewer for appreciating our work and for additional comments. We agree that the comparisons with a disasters could be a major point. For this reason, we added in the text (underlined in yellow) some sentences attempting to explain this complex methodology, both in the introduction and in the discussion. According to reviewer’s suggestions, we tried to clarify how covid-19 created a field of study profoundly different from natural disasters or terrorist attacks, in relation to both definition ( DSM V-IV and ICD-11) and tools used. We added it in the limitations section.
“Quite surprisingly our study showed that high educational level (ISCED > 2) and low 262 monthly income were significantly associated with increased risk of PTSD. This is in con- 263 trast with other studies on trauma exposed adults [58] or after natural disasters [59] and 264 contradicts the theory that education allows to acquire effective tools to face threats and 265 traumatic events [60][61]”
Again, if you compare it to “other disasters”, then you indeed get strange results.
Response: We thank the reviewer for this comment. We extensively reviewed the manuscript and the discussion was re-formulated focusing in two points. Particularly, the second point is newest as stated as following “Secondly, our study could be a further example of the serious methodological issues related to research on PTSD in the context of the COVID-19 pandemic [71]. It is uncertain if DSM-V’s inclusion criteria for PTSD are compatible with the situations that often occur during the COVID-19 pandemic, since not every disease is a life-threatening event, at least for every patient affected [21], [37]. If, as some authors stated [22], adjustment disorder is a more suitable diagnosis because it does not require a stressor that is life-threatening or traumatic [72], it is not surprising that COVID-19 survivors show different risk or protective factors compared to natural disaster or trauma exposed adults”.
“A stay in ICU is itself a traumatic experience, since patients are confronted with their own death, are dependent on machines and might be altered in consciousness, with severely impaired or impossible communication skills [34].” This reference here is not very convincing. It is not a study, but a spotlight, that does not even use references. The author of the reference does not really give a reason for why it might be traumatic – I agree that ICU might be traumatic, but it is not the ICU itself, it is what happens to the person in the ICU
Related to that: how many people mentioned that the ICU was traumatic for them? Because in answering the PTSD scale they needed to mention a specific trauma to which it is related, to fulfill criterion A.
Response: We thank the reviewer for this comment. Although the reference is not very convincing the reviewers, authors believed that the article defines the context of intensive care as at high risk, due to its specificities of care, of generating highly stressful experiences. However, in the subsequent discussion we tried to better clarify what are the potential risk factors for the onset of PTSD, but the literature is not homogeneous in terms of methods and results. In addition, the traumatic experience is not generated by ICU itself, but by the patient’s characteristics and the type, trend and outcome of disease. In our study all patients presenting with PTSD symptoms recall hospitalization as a traumatic experience, but only a fraction of them were able to recall a specific traumatic event related to their ICU stay, probably due to the use of sedatives and early discharge to intermediate-care wards as soon as possible because of ICU beds shortage.
I would really suggest the authors simply compare it to other COVID-19 studies – there are so many.
Response: We thank the reviewer for the comment. We made corrections according reviewer’s suggestion.
In limitations I would add: 1, use of DSM-IV, not the most current use of PTSD; 2, PTSD might not be the best choice of disorder, when compared to adjustment disorders for example.
Response: We thank the reviewer for the comment. We made corrections according reviewer’s suggestion.
Reviewer 2 Report
Dear authors,
Thank you for the interesting proposal. I would like to submit some comments which contents, in my opinion, may upgrade your proposal.
Abstract: please avoid not including the whole name of the psychometric instruments (e.g. EQ-5D-5L). Additionally, the abstract benefits from proposals for some future paths of research, considering the findings highlighted with this paper proposal.
Results: The variable 'age' is very important to be more clearly mentioned. Figure 2 addresses it and we're informed about many people having more than 60 years but main results benefit from making clear what is the mean age of subjects. We do not know if there are many elderly and that is crucial to understand if a neurocognitive instrument for screening cognitive deficits should be considered it this research design, for instance in a presencial additional session.
Study design and participants: Besides the above mentioned aspects, was there any screening question regarding previous psychiatric history? All of the assessed subjects have presented for the first time the PTSD indicators? If so it needs to be clarified, if it wasn't it may be considered an important limitation of the study.
Conclusions: In this section should be addressed future researches proposals regarding the potential influence of psychosocial and psychopathological variables in the quality of life of other similar clinical groups, namely to better recognize the prognosis and the evolution of diseases.
Author Response
Dear authors, Thank you for the interesting proposal. I would like to submit some comments which contents, in my opinion, may upgrade your proposal.
Response: We thank the reviewer for appreciating our work and for additional comments.
Abstract: please avoid not including the whole name of the psychometric instruments (e.g. EQ-5D-5L).
Response: We thank the reviewer for the comment, we added in the abstract the complete name of EQ5D5L.
Additionally, the abstract benefits from proposals for some future paths of research, considering the findings highlighted with this paper proposal.
Response: We thank the reviewer for the comment, I added in the manuscript the sentence “ Future research areas should be oriented towards recognizing potential psychosocial and psychopathological variables capable of influencing the quality of life of patients discharged from intensive care unit to better recognize prognosis and longtime effects of diseases”
Results: The variable 'age' is very important to be more clearly mentioned. Figure 2 addresses it and we're informed about many people having more than 60 years but main results benefit from making clear what is the mean age of subjects. We do not know if there are many elderly and that is crucial to understand if a neurocognitive instrument for screening cognitive deficits should be considered it this research design, for instance in a presencial additional session.
Response: We thank the reviewer for the comment, I added the information about 4 age categories in Table 1. 95 on 142 patients had more than 60 years. However, the difference between the categories is not statistically significant.
Study design and participants: Besides the above mentioned aspects, was there any screening question regarding previous psychiatric history? All of the assessed subjects have presented for the first time the PTSD indicators? If so it needs to be clarified, if it wasn't it may be considered an important limitation of the study.
Response: We thank the reviewer for the comment, It would have been interesting to use a neurocognitive tool for screening cognitive deficits or previous psychiatric history for elderly (>60 years) patients in a presential additional session, but it was not possible at that time due to the social restriction imposed. This is clearly an important limitation, as stated in the paragraph “limitations”.
Conclusions: In this section should be addressed future researches proposals regarding the potential influence of psychosocial and psychopathological variables in the quality of life of other similar clinical groups, namely to better recognize the prognosis and the evolution of diseases.
Response: We thank the reviewer for the comment. In that paragraph we added the sentence: “Future research areas should be oriented towards recognizing potential psychosocial and psychopathological variables capable of influencing the quality of life of patients discharged from intensive care unit to better recognize prognosis and longtime effects of diseases”